DOI: 10.1038/ncomms13333　　OPEN

# Bacterial viruses enable their host to acquire antibiotic resistance genes from neighbouring cells

Jakob Haaber[1], Jørgen J. Leisner[1], Marianne T. Cohn[1,†], Arancha Catalan-Moreno[1,†], Jesper B. Nielsen[2], Henrik Westh[2,3], José R. Penadés[4] & Hanne Ingmer[1]

Prophages are quiescent viruses located in the chromosomes of bacteria. In the human pathogen, *Staphylococcus aureus*, prophages are omnipresent and are believed to be responsible for the spread of some antibiotic resistance genes. Here we demonstrate that release of phages from a subpopulation of *S. aureus* cells enables the intact, prophage-containing population to acquire beneficial genes from competing, phage-susceptible strains present in the same environment. Phage infection kills competitor cells and bits of their DNA are occasionally captured in viral transducing particles. Return of such particles to the prophage-containing population can drive the transfer of genes encoding potentially useful traits such as antibiotic resistance. This process, which can be viewed as 'auto-transduction', allows *S. aureus* to efficiently acquire antibiotic resistance both *in vitro* and in an *in vivo* virulence model (wax moth larvae) and enables it to proliferate under strong antibiotic selection pressure. Our results may help to explain the rapid exchange of antibiotic resistance genes observed in *S. aureus*.

[1] Department of Veterinary Disease Biology, Faculty of Health and Medical Sciences, University of Copenhagen, Stigbøjlen 4, DK-1870 Frederiksberg, Denmark. [2] Department of Clinical Microbiology, Hvidovre University Hospital, DK-2650 Hvidovre, Denmark. [3] Department of Clinical Medicine, Faculty of Health and Medical Sciences, University of Copenhagen, DK-2200 Copenhagen, Denmark. [4] Institute of Infection, Immunity and Inflammation, Glasgow University, Glasgow G12 8TA, UK. † Present addresses: Novozymes, Krogshøjvej 36, DK-2880 Bagsværd, Denmark (M.T.C.); Instituto de Agrobiotecnología (IDAB), Avda de Pamplona 123, 31192 Mutilva, Spain (A.C.-M.). Correspondence and requests for materials should be addressed to H.I. (email: hi@sund.ku.dk).

Staphylococcus aureus causes serious and life-threatening infections in humans, while at the same time colonizing asymptomatically the skin and anterior nares of ~30% of the human population[1]. For S. aureus, antibiotic resistance is a growing healthcare problem. Methicillin resistant S. aureus strains have spread in recent years[2] and resistance to other clinically relevant antibiotics is developing[3]. Antibiotic resistance genes are often carried on mobile genetic elements including staphylococcal cassette chromosomes, S. aureus pathogenicity islands (SaPIs), plasmids, transposons and phages[3], which make up 15–20% of the total S. aureus genome[4]. Antibiotic resistance genes are commonly obtained by horizontal gene transfer of mobile genetic elements, and in S. aureus phages are believed to be particularly important for the transfer[5]. Nearly all strains carry phages stably integrated in the bacterial chromosome as prophages[4,5] and extensive mobility of phages and other mobile genetic elements has been demonstrated between close relatives of S. aureus[6]. The flexibility of a stable core genome and a large accessory mobile genome is believed to account for the extreme adaptability of S. aureus[4,5,7]. For example, such flexibility allowed antibiotic-resistant clones of S. aureus to emerge and dominate in a hospital environment characterized by a high antibiotic selection pressure[8] and in a controlled in vivo study transfer of mobile genetic elements between S. aureus strains was observed just 4 h after colonization of piglets[9].

Phage-mediated transfer of genetic material is known as transduction. Although prophages mostly remain inactive and are replicated with the bacterial chromosome, some cells in prophage-containing (lysogenic) populations may undergo prophage induction. Hereby, the phages enter a lytic life cycle where they replicate, lyse the host cell, spread as phage particles, and infect and lyse phage-susceptible cells in the environment[10]. During replication, phage particles may occasionally encapsulate fragments of bacterial DNA that can be transferred into newly infected cells. This process of transduction, that is, the transfer of genetic material from a donor to a recipient cell through such transducing particles, was one of the first tools employed in molecular biology to transfer genetic material between bacterial cells[11,12]. Although being a useful genetic tool, transduction is generally inefficient as transduced cells may be killed by co-infection of virulent phages present during the transduction process[13]. With this in mind, it is puzzling how phages contribute to the extensive shuffling of genes that occurs among S. aureus strains in vivo[4–9,14].

Here we describe how phages released from a subpopulation of lysogenic S. aureus are instrumental in acquiring antibiotic resistance genes from neighbouring, phage-susceptible bacteria and returning these resistance genes to the remaining, lysogenic population. As the lysogenic S. aureus cells are immune to phage-mediated killing, this process, which we have termed 'auto-transduction', is highly efficient.

## Results

**Bacterial lysogens acquire antibiotic resistance genes**. To study the transfer of antibiotic resistance genes among S. aureus strains, we co-cultured one lysogenic strain with one of three non-lysogenic strains. The lysogenic strain (8325-S) carried three prophages ($\phi$11, $\phi$12 and $\phi$13) and was a derivative of the NCTC 8325 being streptomycin-resistant due to a mutation in rpsL (A302G). The other strains were derivatives of the non-lysogenic strain 8325-4, one of which (designated 8325-4 chrom) was resistant to erythromycin due to ermB inserted in the chromosomal rot gene; a second strain (designated 8325-4 plasmid) was resistant to chloramphenicol due to a cat gene on the non-conjugative plasmid pRMC2; and a third strain (designated 8325-4

SaPI) was resistant to tetracycline due to a tetM gene in the S. aureus pathogenicity island SaPIbov1, a chromosomal genetic element highly mobilized by phage $\phi$11[15]. Phenotypically, the lysogenic strain 8325-S can be distinguished from the non-lysogenic strains by a non-haemolytic phenotype caused by elevated activity of the regulatory proteins SigB and SarS in 8325 strains as compared with the 8325-4 derivatives[16].

After selection of cells resistant to a combination of two antibiotics (streptomycin–tetracycline, streptomycin–chloramphenicol and streptomycin–erythromycin, respectively), we found double resistant mutants at the frequencies of $3 \times 10^{-3}$, $5 \times 10^{-4}$ and $4 \times 10^{-6}$ per total colony-forming units (CFUs) for 8325-4 SaPI, 8325-4 plasmid and 8325-4 chrom, respectively (Fig. 1a). Surprisingly, as with 8325-S, all these cells were non-haemolytic, which suggests that antibiotic resistance genes were transferred from the non-lysogenic strains to the lysogenic strain 8325-S (Fig. 1b). This notion was confirmed when we obtained the same resistance frequencies using a lysogenic strain that was resistant to both streptomycin and rifampicin (termed 8325-SR) and selecting for both of these resistances together with the antibiotic resistance marker being transferred (Fig. 1a and Supplementary Fig. 1). The transfer was not restricted to specific antibiotic resistance genes, plasmids or genomic locations of the resistance markers, as transfer was observed with other antibiotic resistance markers at variable plasmid and genomic locations, and verified by whole genome sequencing (Supplementary Fig. 2). No double- or triple-resistant colonies were observed (detection limit is $< 1e^{-9}$ and $< 9e^{-10}$ CFU ml$^{-1}$, respectively) when 8325-S or 8325-SR was co-cultured with 8325-4 not harbouring an antibiotic resistance marker (Fig. 1a). Further, when the lysogenic strains 8325-S or 8325-SR were substituted with the phage-cured derivatives (JH977 or JH978, respectively) the transfer of resistance genes was abolished, demonstrating that a prophage is essential for the transfer.

**Released phages mediate gene acquisition**. The central role of released phage in recruitment of antibiotic resistance genes was demonstrated by the findings that the transfer required Ca$^{2+}$ (Supplementary Fig. 3 and Fig. 1c), which is chelated by citrate and is needed for adsorption of phages to bacterial cells[13], did not require contact between the lysogenic and the phage susceptible cells (Fig. 1c) (see Methods and Supplementary Fig. 4 for experimental setup) and was insensitive to DNase treatment ruling out natural competence. However, $\phi$11 is capable of transduction[17] and transfer was abolished in strains harbouring either a $\phi$11 derivative defective in packaging DNA into phage particles (terL mutant) and severely reduced (5,000-fold) in a strain expressing a non-inducible CI phage repressor (Fig. 1d). In addition, transfer was dependent on RecA-mediated CI repressor cleavage but not on induction of the SOS response (Supplementary Fig. 5). Collectively, these results demonstrate that released phage $\phi$11 is necessary and sufficient for the lysogenic strain to recruit DNA by transduction. As the release of $\phi$11 phage appears to be a means for the lysogenic population to acquire DNA from neighbouring phage-susceptible target cells, the process may be explained as 'autotransduction'.

To demonstrate the efficacy of autotransduction in comparison with DNA transfer into phage-susceptible cells (the common perception of transduction), we used a sterile filtered $\phi$11 phage lysate obtained from 8325-4 SaPI to infect either phage cured 8325-SR (JH978) or the lysogen, 8325-SR, at multiplicity of infection (MOI) of 0.1 and 1. Such lysates will contain the phage and rare transducing particles carrying DNA from the 8325-4 SaPI strain. Following infection of 8325-SR, transductants were obtained at frequencies proportional to the amount of phage

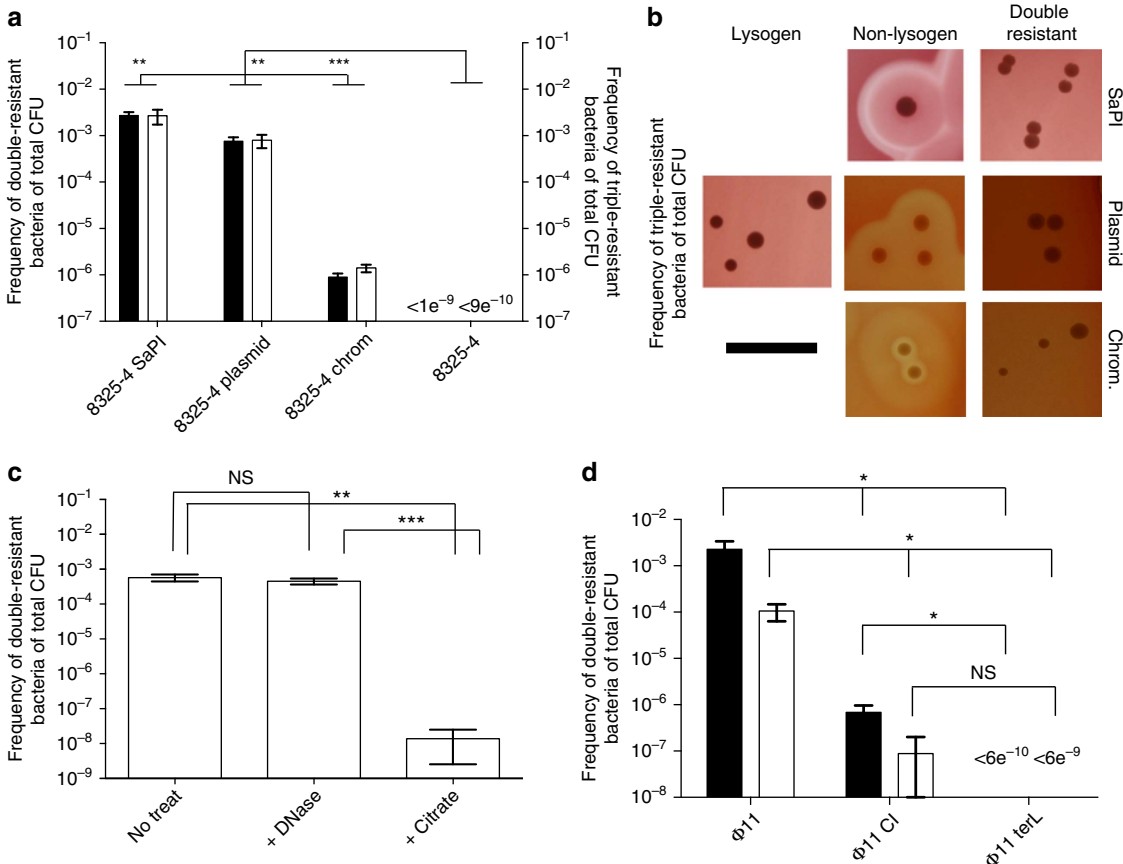

**Figure 1 | Phages transfer DNA from non-lysogenic cells back to the lysogenic host.** (**a**) Frequencies of double or triple resistant cells of total CFU when co-culturing strains 8325-4 SaPI, 8325-4 plasmid, 8325-4 chrom and the non-marker control 8325-4 with 8325-S (filled bars) or 8325-SR (open bars), respectively. No double or triple resistant cells were observed when the lysogen was substituted with the phage-cured derivatives JH977 or JH978 (detection limit = $4e^{-10}$). (**b**) Haemolysis phenotype of parental strains 8325-S (lysogen), 8325-4 (non-lysogen) carrying SaPI, plasmid or chromosomal markers, and double resistant colonies obtained from co-cultures shown in **a** (scale bar, 10 mm). (**c**) Frequencies of double resistant cells in cultures where 8325-S and phage-susceptible 8325-4 SaPI populations were grown separately but sequentially exposed to sterile-filtered culture supernatant and in presence of DNase or citrate, the latter of which binds $Ca^{2+}$ (see Methods and Supplementary Fig. 4 for details of experimental setup). (**d**) Frequencies of double resistant CFU in co-cultures of 8325-4 SaPI (filled bars) or 8325-4 plasmid (open bars) and streptomycin-resistant lysogens harbouring a wild-type phage (Φ11) or derived phage mutants (Φ11 CI and Φ11 terL). For frequency of spontaneous antibiotic resistance development, see Supplementary Table 3. Error bars = s.d., $n = 3$. ***$P < 0.001$,**$P < 0.01$ and *$P < 0.05$, NS, not significant ($t$-test).

lysate added (Fig. 2a,b). In contrast, infection of JH978 led to a dramatic reduction in bacterial numbers and no transductants irrespective of the MOI (Fig. 2a,b). This is likely to be due to the extensive, phage-mediated killing that occurs of both the infected strain and any transductants formed during the 24 h experiment. In traditional transduction experiments, phage exposure is brief and the transductants are saved when adsorption of phage is halted by citrate-mediated chelation of calcium ions[13,18]. In contrast, lysogenic cells already carrying a prophage will be immune to killing by their own phage[19,20] and therefore these restrictions do not apply to autotransduction. The phage lysate was demonstrated to be sterile and in control cultures ($2–4 \times 10^9$ CFU ml$^{-1}$) that had no added phages, no resistant colonies were obtained corresponding to a spontaneous resistance frequency against tetracycline of $< 5 \times 10^{-9}$. Consequently, in the absence of laboratory manipulations, autotransduction of a SaPI-borne antibiotic resistance marker is $\sim 10^6$-fold more efficient than regular transduction (Fig. 2a,b).

The majority of *S. aureus* strains are lysogenic for one or more phages[4] that could potentially interfere with the autotransduction process. To determine whether DNA can be recruited from strains carrying prophages other than the φ11 phage family[6], we examined USA300 and Newman, which are lysogenic with prophages φSA2usa, φSA3usa and φNM1 to φNM4, respectively. In addition, USA300 carries erythromycin and tetracycline resistance markers on native plasmids and Newman carries a plasmid-borne tetracycline marker. Co-culture and subsequent selection for 8325-SR and either of the two resistance markers of USA300 in both cases resulted in a frequency of $\sim 6 \times 10^{-5}$ of triple resistant colonies per total CFU, while a lower frequency of $3 \times 10^{-6}$ was observed in the Newman strain (Supplementary Fig. 6) probably reflecting that phage φ11 infects USA300 more efficiently than Newman as shown in Supplementary Table 1. These results show that autotransduction occurs despite the presence of prophages other than φ11 in the target strains.

**Prophages provide a competitive advantage to its host.** The efficacy of autotransduction is in part attributed to virulent phages propagating on and killing the phage-susceptible target strain (Fig. 2). To assess the competitive advantage of harbouring the φ11 prophage, we co-cultured 8325-SR with 8325-4 chrom (Supplementary Fig. 7a) or USA300 (Supplementary Fig. 7c). Although the growth rate of 8325-SR was lower than those of

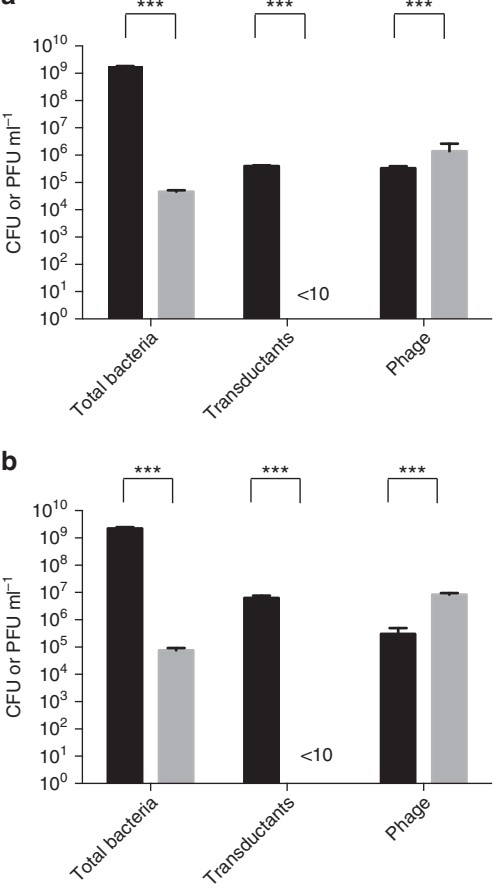

**Figure 2 | Acquisition of foreign DNA by lysogenic and non-lysogenic strains.** Lysogenic 8325-SR (black bars) and non-lysogenic JH978 (grey bars) strains were exposed for 24 h to a phage lysate obtained from 8325-4 SaPI at MOI = 0.1 (**a**) or 1 (**b**) after which the total number of bacteria, transductants and phages were determined by plating without (bacteria) and with antibiotic selection (transductants) or by spotting for phage titers (phage). Error bars = s.d., $n = 3$. \*\*\*$P < 0.001$ (*t*-test).

both the non-lysogenic competitors (Supplementary Table 4), the lysogen efficiently outcompeted the non-lysogenic target cells concomitant with significant production of free phages (Supplementary Fig. 7b,d, respectively), an effect that was greatly reduced by citrate. Within the timeframe of the experiment lysogenized target cells were not detected, indicating that on a short timescale released phages will provide the lysogenic population with a competitive advantage over the phage-susceptible targets, as previously seen[21]. This result supports earlier observations that released prophages may function as biological weapons in competition with other bacteria[22].

Given the dual action of prophages in acquisition of DNA and killing of phage-susceptible cells, we hypothesized that auto-transduction could be advantageous for the lysogen during establishment in a new environment with a pre-existing bacterial population. For this we chose to investigate how lysogenic 8325-SR competed against another phage-susceptible strain (USA300) when this was already growing on the surface of agar plates, by analysing CFU of both strains over time. After 1 day, the initial exponential growth had ceased for both strains and the number of CFUs gradually declined over the next 16 days. Although the lysogen 8325-SR experienced a 200-fold reduction during the 16 days of the experiment, the USA300 population experienced a 5,000-fold decline in the presence of 8325-SR (Fig. 3a).

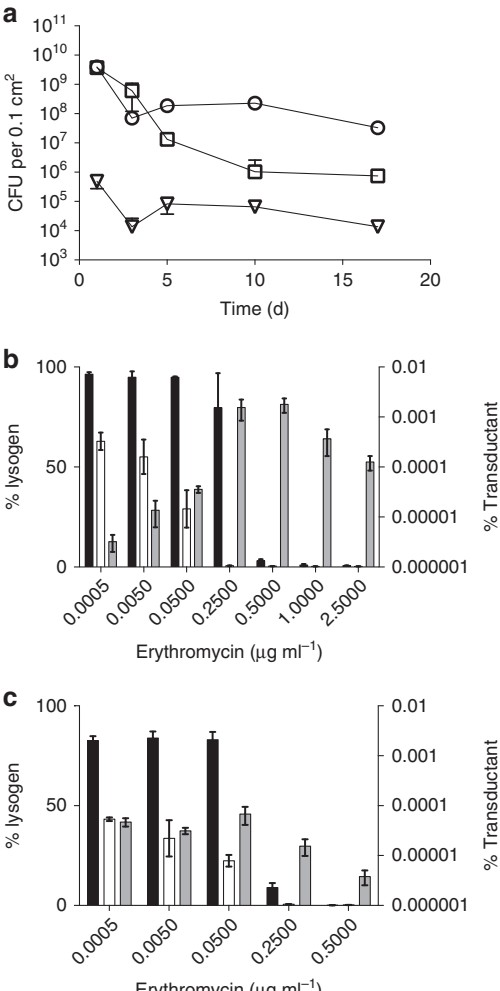

**Figure 3 | Autotransduction occurs without and with antibiotic selection.** (**a**) At time zero, lysogenic 8325-SR (circles) was spotted on agar plates without antibiotics containing a growing USA300 population (squares) forming autotransduced 8325-SR (triangles). At selected time points, equal size areas of the agar were stamped out, cells extracted and plated on blood agar plates containing the relevant antibiotics. (**b,c**) Percentage lysogen (8325-SR) in liquid co-cultures with phage susceptible but erythromycin-resistant 8325-4 chrom (**b**) or USA300 (**c**) grown in the presence of erythromycin with (open bars) or without citrate (black bars). Percentage autotransductants of total population in the absence on citrate (grey bars) plotted on log-scale axis. No autotransductants were obtained in cultures with citrate. Error bars = s.d., $n = 3$.

Importantly, even though no antibiotic selection was applied in the agar plates, we observed a low number of 8325-SR colonies resistant to tetracycline throughout the experiment, indicating that the *tetK* marker initially located on a USA300-borne plasmid was transferred to and maintained in 8325-SR without antibiotic selection (Fig. 3a).

To assess the impact of autotransduction during antibiotic selection, we co-cultured the erythromycin-sensitive lysogen 8325-SR together with the erythromycin-resistant 8325-4 chrom (Fig. 3b) or USA300 (Fig. 3c) in the presence of increasing concentrations of erythromycin. Even at antibiotic concentrations lethal to 8325-SR, erythromycin-resistant 8325-SR transductants emerged in the co-cultures, while the original lysogenic population gradually declined with increasing antibiotic concentrations. This effect was abolished by citrate. These data show that

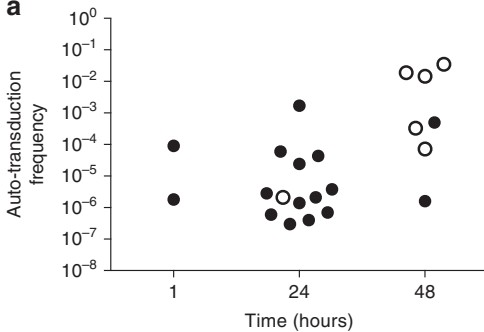

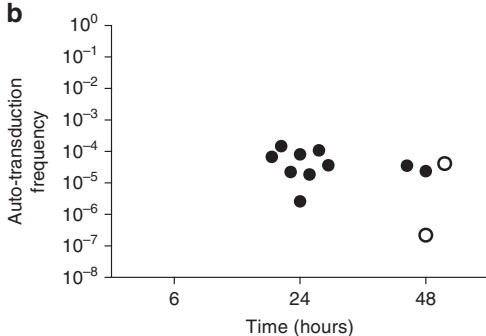

**Figure 4 | Autotransduction occurs in an *in vivo* infection model.**
Autotransduction frequencies recorded over time during co-infection of *G. mellonella* larvae with 8325-SR and (**a**) 8325-4 SaPI or (**b**) USA300. At each time point, 20 larvae were sampled and total CFU and triple-resistant colonies were determined from the haemolymph. The calculated autotransduction frequencies are shown for the larvae in which autotransduction was observed. Filled and open symbols represent living and dead larvae, respectively. No autotransduction was observed in the inoculum used for infection; when lysogenic bacteria were substituted with a phage-cured derivative (JH978); when the non-lysogenic strain was substituted with 8325-4 not carrying an antibiotic resistance marker or when larvae were injected with a PBS control (detection limit = 2e−9).

autotransduction allows the lysogen to survive and increase in numbers even in the presence of counter selection, a trait that is not possible when the phage is inactivated.

Lastly, to examine whether autotransduction occurs outside the test tube, we infected *Galleria mellonella* wax moth larvae, a commonly used virulence model for *S. aureus* with a 1:1 mix of 8325-SR and either 8325-4 SaPI or USA300. Just 1 hour post infection, autotransduction was observed in the larvae infected with the mix containing 8325-4 SaPI (Fig. 4a) but not after 6 h in the mix containing USA300 (Fig. 4b) and not in the inoculum used for injection. At 24 h post infection, autotransduction approached frequencies comparable to those observed in laboratory medium both in larvae infected with 8325-SR and 8325-4 SaPI or USA300. No transduction of 8325-4 SaPI or USA300 was observed, implicating autotransduction in the efficient spreading of antibiotic resistance genes *in vivo*.

## Discussion

Here we describe a process by which spontaneously released phages from a subpopulation of lysogenic bacteria propagate on phage-susceptible, co-cultured target bacteria and, subsequently, with high frequencies, transfer DNA from the lysed cells back to the remaining lysogenic cell population. Owing to the activity of the phage repressor, phages are unable to kill cells already carrying the prophage[19,23] but injection of DNA is not inhibited

and this allows the rare, transducing particles to deliver their content to the immune, lysogenic cells where it may be incorporated (Fig. 5). Because of its resemblance to transduction, we have termed this process 'autotransduction'.

In *S. aureus*, the process of autotransduction by φ11 is highly efficient compared with transduction of cells that are not lysogenic for the incoming phage. We found that $10^6$-fold more lysogenic cells are able to receive and establish an antibiotic resistance marker present on a SaPI element compared with cells that are not lysogenic. In fact, transduction of non-lysogenic cells could only be detected if citrate was used to prevent phage-mediated killing of transductants. Similarly, autotransduction, but not transduction of non-lysogenic cells, took place in the wax moth larva *G. mellonella*. These findings suggest that in natural environments resistance genes may be transferred by auto-transduction.

*S. aureus* φ11 belongs to serological group B[24] and generally phages from serological groups B and F are capable of transduction[25]. More recently, the transducing ability of phages from serological groups A[26] and L[27] has been characterized and other accounts of transducing phages have been reported[7,28], indicating that transduction capability is widespread among staphylococcal phages. The host range of staphylococcal phages is generally limited[6,20], indicating that autotransduction is restricted to occur between closely related strains. However, phages can mediate the extensive transfer of genetic material between such closely related *S. aureus* strains during *in vivo* colonization and this adaptability is key to *S. aureus* success as a colonizer[5–8]. In a recent study, DNA mobility was followed between pig-colonizing *S. aureus* strains, where transfer of mobile genetic elements was observed after just a few hours[9]. We show that auto-transduction occurs at high frequency, allows lysogenic cells to proliferate under conditions that are otherwise lethal for this population, and that the acquired DNA is not restricted to specific antibiotic resistance genes, plasmids or chromosomal locations. Overall, this suggests that autotransduction may contribute to the mobility of genetic elements between *S. aureus* genomes[26,29,30] and as such may play a role in successful colonization and infection by the organism. For autotransduction to take place, the phage must be a generalized transducer and the lysogen susceptible to superinfection. However, if only a fraction of the world's $10^{32}$ estimated phages harbour these properties, autotransduction may be a major mechanism in bacterial acquisition of DNA. In fact, Zinder and Lederberg[31] already noted in their pioneering work on transduction of *Salmonella* that transducing particles could enter lysogenic cells. Although the impact of this observation on efficacy of transduction in natural environments was not pursued, it shows that autotransduction is probably not restricted to *S. aureus*.

The autotransduction process requires that a fraction of the lysogenic cells are killed, to release the phages. Although the phage release appears as an altruistic act in which lysis of a subpopulation benefits the remaining intact lysogenic population, it can also be viewed as spite, as the phage release causes lysis of the competing target cells[32]. In this respect, autotransduction resembles some colicinogenic systems where individual colicin-producing and lysing cells contribute to the fitness of the remaining isogenic population[33] but with the essential additional benefit of acquiring genetic material from killed competitors. Alternatively, autotransduction may be viewed as a consequence of 'selfish' phages that improve survival of their bacterial hosts, first by eliminating non-lysogenic competitors and, second, by the acquisition of genes encoding beneficial phenotypes. Further studies are required to detail the respective ecological benefits of the autotransduction mechanism for the prophage and the bacterial host.

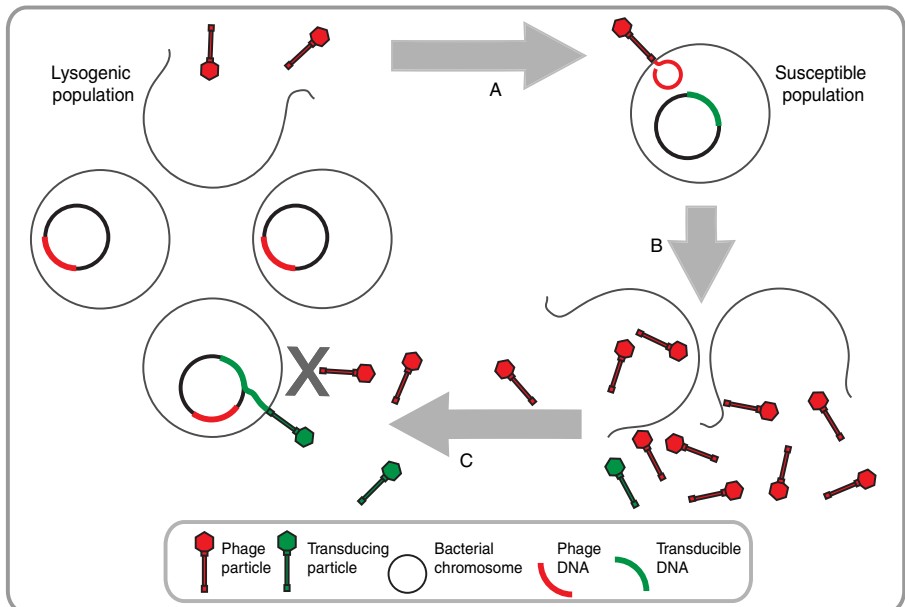

**Figure 5 | Autotransduction model.** (A) Spontaneous phage induction mediated by RecA cleavage of the CI phage repressor from a subpopulation of a lysogenic bacterial population allows the phage to infect a susceptible co-existing bacterial population. (B) Propagation on the susceptible population and subsequent lysis releases phage progeny along with low numbers of transducing particles. (C) Owing to immunity to its own phages, the lysogenic population can 'filter out' the infective phage progeny and allow the occasional transducing particle to deliver its DNA to the lysogenic population by autotransduction.

Our results support the growing evidence of a symbiotic relationship between prophages and their hosts[34,35]. Examples include spontaneous release of phages used as biological weapons to kill competitors[22], the protection by prophages against infection by related phages[36,37], and the induction and expression[38] of phage-carried bacterial virulence genes[39]. However, to our knowledge, it has not previously been demonstrated that phages released from their lysogenic host can act in the interest of the hosting bacterial population by diversifying its genetic repertoire.

## Methods

**Bacterial growth.** Bacterial strains used in the study are listed in Supplementary Table 2. *S. aureus* strains were grown in tryptic soy broth (TSB) from Difco, unless otherwise noted. The medium was supplemented with 10 mM $CaCl_2$ or 20 mM sodium citrate when phage propagation was wanted and unwanted, respectively. Bacterial cultures were supplemented with relevant antibiotics at the following concentrations, unless otherwise noted: $5\,\mu g\,ml^{-1}$ of erythromycin, $10\,\mu g\,ml^{-1}$ of chloramphenicol, $50\,\mu g\,ml^{-1}$ of streptomycin, $0.5\,\mu g\,ml^{-1}$ of rifampicin, $5\,\mu g\,ml^{-1}$ of tetracycline and $100\,\mu g\,ml^{-1}$ of spectinomycin, all purchased from Sigma. Cultures were incubated in 5 ml medium in sterile 15 ml plastic centrifuge tubes at 37 °C with shake (180 r.p.m.).

**Co-culture experiments in liquid.** Relevant strains were inoculated to final $OD_{600} = 0.01$ in mixed cultures without antibiotics. Cultures were incubated for 24 h, unless otherwise noted. Before inoculation and at each sampling, the culture was sonicated as previously described[40], to ensure single-cell distribution in the culture. Enumeration of individual strains in mixed cultures was done by plating dilutions on blood agar plates containing relevant antibiotics. Differentiation between strains in mixed cultures was further aided by differences in haemolysis phenotypes.

Autotransduction frequencies were determined as the fraction of cells in a co-culture that were resistant to the antibiotic markers carried by both of the co-cultured strains relative to the total number of CFU.

Competitive interactions between two co-cultured strains were determined by assessing the percentage of the lysogen relative to the sum of lysogenic and susceptible cells and reported as '% lysogen'.

**Separated culture experimental setup.** To assess whether cell-to-cell contact is needed for DNA transfer, we used cell-free supernatant of a 8325-S culture

containing spontaneously released phages for infection of phage-susceptible 8325-4 SaPI. After 2 h incubation allowing time for lysis of the 8225-4 SaPI population, cell-free supernatant from this culture was used to re-infect the lysogenic 8325-S strain in the presence or absence of citrate or DNAaseI treatment. The DNAaseI activity was demonstrated by comparison of gel electrophoresis and transduction data for DNA with or without prior DNaseI treatment (Supplementary Fig. 4b,c, respectively). After overnight incubation, frequency of double resistant mutants of total CFU was determined.

**Whole genome sequencing and single-nucleotide polymorphism analysis.** DNA from overnight cultures was extracted using DNeasy Blood and Tissue Kit (Qiagen, Hilden, Germany). DNA concentrations were measured and normalized using a Qubit fluorometer (Invitrogen, UK) and libraries were prepared using Illumina Nextera XT DNA preparation kit. Sequencing of libraries was performed on an Illumina MiSeq instrument (Illumina, USA) at the Department of Clinical Microbiology, Hvidovre University Hospital, Denmark. Samples were multiplexed to 24 samples per sequencing reaction and sequenced using $2 \times 150$ bp paired-end reads. Genomes were assembled *de novo* using Velvet (v.1.0.11) and VelvetOptimiser (v.2.1.7). Variants were detected using a reference-based single-nucleotide polymorphism mapping via Stampy[41]. Reads were mapped to the *S. aureus* USA300 strain TCH1516 (Genbank accession no. NC_010079) using Stampy (v.1.0.11) and single-nucleotide polymorphism calling was performed using SAMtools v.0.1.12.

Phylogeny analysis was performed by neighbour-joining analysis based on the pairwise sequence alignment of genomes defined by the reference genome USA300 strain TCH1516. Trees were subsequently created using FigTree (http://tree.bio.ed.ac.uk/software/figtree) and the phylogeny analysis was used to confirm that transductants originated from the lysogenic population.

**Enumeration of phages and efficiency of plaquing.** $\phi 11$ was enumerated by spotting dilutions of 0.2 μm filtered supernatant on the indicator strain RN4220. Other phages released from USA300 or Newman were enumerated by flow cytometry (Becton Dickinson FACS Calibur) as previously described[42]. Following changes to the protocol were applied: lysate was incubated at room temperature instead of 80 °C. Settings were as follows: forward scatter = 404V, side scatter = 605V, green fluorescent channel = 520V. Efficiency of plaquing for lysates from all three strains was calculated as the ratio of plaque-forming units (PFU) to number of phage particles determined by flowcytometry.

**Infection with phage lysate.** A phage lysate was produced by co-culturing 8325-SR with 8325-4 SaPI for 4 h followed by centrifugation and sterile filtration of the supernatant. The titre of the lysate was determined by plaquing on RN4220. The

lysate was used to infect a culture of 8325-SR or 8325-4 (JH978), both at $OD_{600} = 0.01$ at MOI of 1 and 0.1. Following incubation overnight, the total CFU ml$^{-1}$, transductants ml$^{-1}$ and PFU ml$^{-1}$ were determined by plating on appropriate medium.

**Competition on solid surface.** Pre-culture of USA300 was adjusted to $OD_{600} = 0.5$ and 100 μl was spread on agar plates containing 25 mM CaCl$_2$. Following 6 h incubation at 37 °C growth was visible as a thin layer of bacteria. At this time, 10 μl of 8325-SR adjusted to $OD_{600} = 1$ was spotted on to the USA300 bacterial lawn at a ratio of ∼1 cell of 8325-SR per 10 cells of USA300. At selected time points, equal size areas of the agar were stamped out from the middle of the interaction zone and added to 1 ml of 0.9% NaCl. Vigorous vortex mixing and sonication was used to suspend the cells from the agar plug. Dilutions were then spotted on blood agar plates containing the relevant antibiotics to distinguish the 8325-SR, USA300 and transductant populations.

**Competition in liquid culture.** The lysogenic strain (8325-SR) and a phage susceptible strain (either 8325-4 chrom or USA300) were inoculated at 1:1 ratios to $OD_{600} = 0.01$ in TSB containing varying concentrations of erythromycin against which 8325-SR is sensitive (minimal inhibitory concentration = 0.625 μg ml$^{-1}$) and the phage susceptible strains are resistant (minimal inhibitory concentration > 40 μg ml$^{-1}$). Following 24 h incubation the cultures were plated on agar containing antibiotics selective for the lysogen or the susceptible strain, or triple resistant transductant cells, and data were plotted as percentage of the lysogen or transductants relative to the total CFU ml$^{-1}$.

**G. mellonella model.** We used an existing infection model[43,44] modified by reducing the inoculum of bacteria to prevent rapid killing of the larvae[45]. No statistical methods were used to determine sample size. Healthy fifth instar wax moth larvae were randomly chosen from a batch purchased at a local pet store (Minizoo, Tårnby). Using a Hamilton syringe and multi-dispenser, to ensure repetitive injections of 10 μl, 20 larvae per condition were injected with 10 μl S. aureus culture containing a total of 10$^6$ CFU consisting of a 1:1 ratio of 8325-SR and 8325-4 SaPI or USA300, and incubated at 37 °C. After 0, 1 or 6 and 24 h, dead larvae were recorded and haemolymph from all 20 larvae (living and dead) was squeezed into 0.9% NaCl and sonicated before dilutions were spotted on blood agar plates containing relevant antibiotics. Blinding was not done during these experiments.

**Selection of streptomycin- and rifampicin-resistant mutants.** Overnight cultures of relevant strains were adjusted to $OD_{600} = 2$ and 50 μl aliquots were spread on agar plates containing either 100 μg ml$^{-1}$ streptomycin or 0.5 μg ml$^{-1}$ of rifampicin. Colonies appearing after overnight incubation at 37 °C were re-streaked on either streptomycin or rifampicin. The obtained mutants were stored at −80 °C. Growth rate measurements of the resistant mutants compared with the wild-type strains confirmed that no growth defect was associated with the mutations (Supplementary Table 4).

**Transduction.** Standard phage transduction methods using φ11 or φ80α were used to transfer mutations or plasmids of interest between strains[18]. Briefly, phages were propagated on a donor strain containing the plasmid or gene of interest, resulting in a phage lysate containing transducing particles. This lysate was used to infect a recipient strain and transductants were obtained by plating on selective medium. Phage killing of the recipient strain was inhibited by administration of citrate after phage infection.

**Gene expression.** Relevant strains harbouring a recA promoter fused lacZ reporter were inoculated to $OD_{600} = 0.01$ in pre-warmed TSB and incubated at 37 °C. At indicated time points, bacterial growth was monitored using optical density measurement and plate spreading, phage release was monitored by measuring PFU in the supernatant and recA expression was measured using standard β-galactosidase assay[46] with the exception that cells were opened by 30 min exposure to 15 μg ml$^{-1}$ lysostaphin (sigma) at 37 °C.

**Statistics.** All experiments were performed in biological triplicates. Averages were compared by t-test, two-sided, using the GraphPad Prism software.

**Data availability.** All relevant data are available from the authors. Genome sequences have been deposited at European Nucleotide Archive, accession number PRJEB15200.

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

## Acknowledgements

We thank Mathias Middelboe (University of Copenhagen) for assistance with flow cytometry; Kurt Fuursted (Statens Serum Institute) for help with the *G. mellonella* model; Vi Phuong Thi Nguyen (University of Copenhagen) for assistance in the lab; G. Lindahl (University of Copenhagen), Karin Hammer, Mogens Kilstrup (Technical University of Denmark) and Peter Vermij (Bird Eye communications) for comments and suggestions on the manuscript; Alexander Horswill (University of Iowa) for bacterial strains; Peter Kjelgaard (Lund) for constructing strain 8325-4 SaPI; and Tim Evison (University of Copenhagen) for editing of figures and proof reading. The work was funded by a Sapere Aude post doctoral grant (#12-126289) for J.H. and a Danish council for independent research grant 12-127417, as well as Danish National Research Foundation's Centre of Excellence Bacterial Stress Response and Persistence (grant identifier DNRF120) for H.I.

## Author contributions

J.H., J.J.L., M.T.C. and H.I. designed the study. J.H. did the experiments, except for the *G. mellonella* experiments performed by J.H. and A.C-M., and whole genome sequencing performed and analysed by J.B.N. and H.W. J.H., J.J.L. and H.I. analysed the data. J.H., J.J.L., J.R.P. and H.I. wrote the manuscript. All authors approved the final manuscript.

## Additional information

**Competing financial interests:** The authors declare no competing financial interests.

