## [Peer Review File · Nature Communications]

Reviewers' comments:

Reviewer #1 (Remarks to the Author):

A. Summary of the key results

Haaber and colleagues demonstrate that a subpopulation of a lysogenic *S. aureus* undergoes prophage induction and releases phages. These phages infect a population of adjacent cells and occasionally may package and transduce new genes, including antibiotic resistance genes, back to the lysogenic population. They also show that the released prophage helps in competing against another strain, probably by lysis of that strain. In addition, the lysogenic state protects the remaining population from the released phages by superinfection exclusion. The phenomenon, termed "auto-transduction" occurs even under antibiotic-selection pressure and also in an in vivo model.

B. Originality and interest: if not novel, please give references

The study is extremely interesting. Auto-transduction highlights a novel aspect of prophage interactions with their hosts, and to the best of my knowledge has not been yet reported. The advantage of having prophages in competing against other populations is not novel, as referenced by the authors.

C. Data & methodology: validity of approach, quality of data, quality of presentation

The data is overall convincing, and the approach used is valid. However, some flaws require correction in data presentation. These flaws are majorly in lack of appropriate controls, as listed below:

1. Figure 1a - a control "recipient" strain lacking a lysogen should be presented to demonstrate that the lysogen is required for the observed gain of antibiotic resistance. Although this is later shown in a different way, it must also be added to these sets of experiment.
2. Figure 1a - a control "donor" strain lacking antibiotic markers must be presented to account for the level of spontaneous resistance formation (the supplementary figure does not suffice - should be carried out in the same set of experiments).
3. Figure 1b - The entire set of mutants used in Fig. 1a should be presented with regard to their hemolytic properties.
4. Figure 1c - DNase treatment without a control to show that the DNase works is worthless. A control showing a treated versus untreated DNA used for e.g. plasmid transformation must be shown to demonstrate that the DNase indeed prevents transformation.
5. Figure 2a/b - in standard transduction assays one uses a "no-bacteria" and "no phage" controls to show that the phage lysate is indeed sterile (as it was generated in a resistant host) and to determine the spontaneous resistance rate of the recipient strain. These two controls must be shown in these experiments.
6. Figure 3a - to determine the effect of a prophage on the competition ability of a lysogen, one must change only one parameter - with and without a prophage. In this experiment, a lysogen is compared to a different strain (non-isogenic), that is different in tens (or hundreds) of other parameters that may account for the different competitiveness ability. This experiment must be repeated with a lysogenless isogenic strain. The antibiotic markers should also be tested reciprocally in both strains as occasionally they may account for a competitive advantage.
7. Fig. 3c - legend should be rewritten. Presented results are hard to follow.

8. Fig. 4 and the relevant Result section do not considerably add to the manuscript. The authors may considered transferring them to the Supplementary material if a shortened version is required.

9. Figure 5 - again, controls are missing - lysogen-free bacteria should be added as a recipient in one experiment and also an antibiotic-free bacteria should serve as the donor. These controls must be shown to account for exogenous supply of antibiotic resistance markers from the animal flora, or phages from the animal flora etc.

10. Figure 5 - the legend is unclear. What does it mean n= variable? Which larvae were picked? Which were not? It seems that subjective bias is unavoidable in these experiments.

11. Figure 5 - perhaps the most interesting finding in vivo would be the auto-transduction observed from interactions of the lysogen with the exogenous flora. The experiment as carried out now shows results in larvae that entirely mimic the in vitro results, and is thus not highly significant.

D. Appropriate use of statistics and treatment of uncertainties

Statistics is not used to indicate p values. These should be critical in Figure 3, and recommended in other figures as well.

E. Conclusions: robustness, validity, reliability

Conclusions are valid, robust and reliable, provided that the appropriate controls suggested above are implemented and show the expected outcome.

F. Suggested improvements: experiments, data for possible revision

Apart from the above critical points, the experiments are designed to address the issue and provide sufficient ground for the novel conclusions.

G. References: appropriate credit to previous work?

Appropriate credit is given to previous work.

H. Clarity and context: lucidity of abstract/summary, appropriateness of abstract, introduction and conclusions

The manuscript is overall well written and clear. A few comments are listed below.

1. Line 28 - delete "Very".

2. Lines 64-68, and also 248-254 - the authors term the prophage induction as an altruistic behavior of a subpopulation of the cells. Nevertheless, from the prophage point of view - this is a way to expand its progeny by infecting a new competitor host. Therefore, perhaps the simpler way to evolutionary explain the phenomenon is that the prophage gains advantage rather than the subpopulation of cells is behaving altruistically (a less preferred route of evolution).

3. Fig. 2 a/b - It is unclear why no transductants are observed in the lysogen-less strain. The frequency of transduction to the lysogenic strain is $\sim 0.1\%$. One would therefore expect this ratio also in the non-lysogenic strain. The authors indicate that probably no transductants are observed because the non-lysogens are killed, yet show that $\sim 100,000$ of them are not killed. This population should yield ~ 100 transductants. The fact that we do not see it should be explained (perhaps the surviving bacteria are adsorption mutants and therefore cannot be transduced?).

4. Line 155 - rewrite unclear sentence.

5. Lines 160-174 - novelty of the results described in this paragraph is questionable. If manuscript requires shortening - I would recommend transferring it to the supplementary material.

6. Line 192 - remove comma after "sensitive".

Reviewer #2 (Remarks to the Author):

This paper reports an interesting finding that builds on decades of phage work: under some conditions, there are some temperate phages (those that can lysogenize their host) that can transduce foreign DNA into a sensitive host, enabling the rapid acquisition of drug resistance (for example). It is generally thought that because transducing particles are known to be minor constituents of a phage population that transduction is a rare event, and because detection of successful transduction usually requires that the phage carrying foreign DNA infect the host largely in isolation of other phages (so that the host cell is not killed).

What they found is that, for lysogens of one temperate phage in *Staph aureus*, the cells readily pick up foreign DNA in a 3 step process: (a) Lysogens 'induce' at some rate, releasing infective phages, which (b) infect sensitive bacteria in their environment. Phage growth and cell lysis releases new phages, some of which had acquired DNA of that sensitive host and are transducing particles. (c) The new phages now infect the original lysogenic strain, which is immune to viable phage infection but is capable of recombining transducing DNA into its chromosome, thereby changing the genotype. It is (c) that is new, albeit primarily in the context of steps of (a) and (b). The study thus provides an interesting mechanism for how transduction can be efficiently observed.

There are a few qualifications that limit the generality of this mechanism. The paper does not delve into any of them.

1) The phage must not only be temperate, but it must be a generalized transducer. A requirement for this is that the phage uses a headful packaging mechanism, which allows non-specific DNA to be packaged into new phage particles. Perhaps half of all phages do not use this mechanism.

2) The lysogen must be susceptible to superinfection, requiring that the receptors are unaltered by lysogenic conversion and that no superinfection exclusion process is conferred by the prophage. Many temperate phages do not have these properties, although the primary phage (ϕ 11) considered in the study is appropriate.

3) The lysogen must sometimes be in an environment with other hosts that are sensitive to the phage (and not already lysogens for that phage). Thus, this mechanism is not likely one for DNA transfer across broad taxonomic levels.

The manuscript is very long for its content. In no small part this is because of the simplistic and uneven descriptions of both the formation of transducing particles and of the transduction step. These are both well-established and widely appreciated processes and the text should better reflect what is known and not present material in a classroom style that is more appropriate for beginning students. The manuscript would also benefit from a better use of original references to specific points or of authoritative reviews rather than the plethora of secondary or derivative publications currently cited.

Throughout the manuscript the authors must clearly distinguish the efficiency of autotransduction and transduction from the efficiency of observing or detecting transductants - e.g., only the latter is 10^6 more efficient (line 142).

Minor comments:

19. replace "dormant" with "quiescent"

25. "so-called" is a completely inappropriate term

75. are phi12 and phi13 also capable of transduction? Do they affect any of the data shown (it is appreciated that some experiments were done with phi11 alone).

244-5. P22, the phage used by Zinder and Lederberg, is now understood to confer both superinfection exclusion and lysogenic conversion when present as a prophage.

Reviewer #3 (Remarks to the Author):

Introduction

The manuscript by Haaber et al. presents data supporting the conclusion that release of phage from bacterial cells allows viable prophage-containing cells to acquire antibiotic-resistance genes released from phage-susceptible strains. The authors term this process 'auto-transduction' and suggest that this process could explain why *S. aureus* can rapidly acquire antibiotic resistance.

General Points for Consideration

The work is of interest to those working in the area of antibiotic resistance and represents a novel finding of significance. The data is of a high standard and the appropriate statistics have been applied. However, the manuscript is difficult to read and understand. The authors would be advised to re-write the Results section to facilitate better understanding by researchers that are not familiar with the concepts of phage biology. Also, the Figure legends are poor and should include more and better explanation of the presented data.

It would be beneficial to the reader if the Supplementary Table 2, listing the strains that were used, was in fact included in the actual manuscript to aid reader understanding.

Specific Points for Consideration

In all co-culture experiments with different *S. aureus* strains there appears to be no consideration of the possibility that different strains could have different growth rates. If this were the case have the authors measured this? In competition experiments how would different growth rates resulting in potentially widely different proportions of the different strains effect results in terms of frequency of resistance, CFU or auto-transduction? This must be addressed.

Throughout the manuscript relative cell numbers are expressed as OD600 values. This is an inaccurate measure of cell numbers. The size of bacterial populations should be expressed as CFU/mL.

Line 36 - the statement that MRSA is spreading epidemically is overstated. For example, incidence of MRSA in UK hospitals is in decline over the past few years due to effective interventions to break the chain of infection.

Line 75 - explain the significance to the reader of the mutation in rpsL.

Line 82 - the lysogenic strain can be distinguished from the non-lysogen because it is non-haemolytic due to increased activity of SigB and SarS. Referring to the previous comment above could the differential activity of these important regulatory proteins effect growth rate and thus

results in competition experiments? could they effect intrinsic antibiotic resistance? etc. Has this been measured?

Line 85-100 needs to be rewritten. It is difficult to understand and needs to link with Figure 1 more effectively. For example, the reader is referred to Fig 1a in line 89 but there is no explanation of what strain 8325-SR is until line 93.

In Figure 1, why is the frequency of double-resistant bacteria identical to triple-resistant bacteria for both 8325-S and 8325-SR? Explain.

Line 100 - after the sentence insert data not shown.

Line 283 - This sentence must be re-written for understanding.

Line 317 - The sentence beginning 'In the absence..' needs to be re-written with a full explanation for understanding.

Line 320 - define abbreviations FSC, SSC.

The data presented in Figure 5 is inadequately presented and explained. The authors state that a 'lower inoculum' was used to prevent rapid killing. They then state that an inoculum of $10E6$ was used - this seems high? The authors need to present the number of larvae that died out of the 20 inoculated in Fig 5a and b. In Fig 5a there are a total of 19 data points from each larva suggesting only 1 died? In Fig 5b there are 10 suggesting 10 died? Because the authors only sampled living larvae would this disparity in the number of surviving larvae not skew the auto-transduction calculation - in fact, why not sample bacteria from all infected larvae alive or dead?

Line 359 - how was the 'multi-dispenser' used with the Hamilton syringe to infect larvae? Explain.

Point by point response to reviewers comments:

Below please find our response to the reviewers' comments:

Response to reviewer #1:

A. Summary of the key results

Haaber and colleagues demonstrate that a subpopulation of a lysogenic *S. aureus* undergoes prophage induction and releases phages. These phages infect a population of adjacent cells and occasionally may package and transduce new genes, including antibiotic resistance genes, back to the lysogenic population. They also show that the released prophage helps in competing against another strain, probably by lysis of that strain. In addition, the lysogenic state protects the remaining population from the released phages by superinfection exclusion. The phenomenon, termed "auto-transduction" occurs even under antibiotic-selection pressure and also in an in vivo model.

B. Originality and interest: if not novel, please give references

The study is extremely interesting. Auto-transduction highlights a novel aspect of prophage interactions with their hosts, and to the best of my knowledge has not been yet reported. The advantage of having prophages in competing against other populations is not novel, as referenced by the authors.

C. Data & methodology: validity of approach, quality of data, quality of presentation

The data is overall convincing, and the approach used is valid. However, some flaws require correction in data presentation. These flaws are majorly in lack of appropriate controls, as listed below:

1. Figure 1a - a control "recipient" strain lacking a lysogen should be presented to demonstrate that the lysogen is required for the observed gain of antibiotic resistance. Although this is later shown in a different way, it must also be added to these sets of experiment.

Response: The experiment for which the result is shown in fig 1a was repeated and phage-cured derivatives of 8325-S (designated JH977) and 8325-SR (designated JH978) were included. When using these strains the transfer of resistance was completely abolished demonstrating that lysogeny is required in order to gain antibiotic resistance (data indicated in text, page 5, line 102-104 and in legend to figure 1a).

2. Figure 1a - a control "donor" strain lacking antibiotic markers must be presented to account for the level of spontaneous resistance formation (the supplementary figure does not suffice - should be carried out in the same set of experiments).

Response: When repeating the experiment reported in figure 1a we also included a donor strain lacking an antibiotic resistance marker (8325-4 in figure 1a) and with this strain we did not observe any double- or triple resistant cells. These data are added as

the last column in fig 1a and mentioned in the text line page 5, 99-102.

3. Figure 1b - The entire set of mutants used in Fig. 1a should be presented with regard to their hemolytic properties.

Response: We have presented the entire set of bacteria from the co-culture experiment reported in fig. 1a using both the lysogen, 8325-S (new fig. 1b) and 8325-SR (new supplementary figure 1).

4. Figure 1c - DNase treatment without a control to show that the DNase works is worthless. A control showing a treated versus untreated DNA used for e.g. plasmid transformation must be shown to demonstrate that the DNase indeed prevents transformation.

Response: The activity of the exact same preparation of DNase was confirmed by comparing transformation frequency of DNA before and after DNaseI treatment (please, see Materials and Methods, page 15, line 296-298 and supplementary fig. 4b and c).

5. Figure 2a/b - in standard transduction assays one uses a "no-bacteria" and "no phage" controls to show that the phage lysate is indeed sterile (as it was generated in a resistant host) and to determine the spontaneous resistance rate of the recipient strain. These two controls must be shown in these experiments.

Response: These controls were included in the original experiments but not mentioned in the manuscript when originally submitted. The phage lysate was demonstrated to be sterile when plated undiluted on non-selective plates. When the "no-phage" controls were plated on non-selective plates we obtained $2-4 \times 10^9$ cfu/ml for the two strains and when plated on selective plates we did not obtain resistant mutants corresponding to a spontaneous resistance frequency against tetracycline of $<5 \times 10^{-9}$. Both findings have been added to the text, page 7, line 136-139.

6. Figure 3a - to determine the effect of a prophage on the competition ability of a lysogen, one must change only one parameter - with and without a prophage. In this experiment, a lysogen is compared to a different strain (non-isogenic), that is different in tens (or hundreds) of other parameters that may account for the different competitiveness ability. This experiment must be repeated with a lysogenless isogenic strain. The antibiotic markers should also be tested reciprocally in both strains as occasionally they may account for a competitive advantage.

Response: In addition to the comment made here, this reviewer points out in point 5 below that there is limited novelty in the original manuscript line 160-174, which was the description of the results reported in figure 3a and 3b. The experiment reported demonstrates the ability of phage released from a lysogen to kill a co-cultured, phage susceptible strain. Since we agree with the reviewer that this is basic phage biology and not central to the manuscript and since reviewer 2 finds the manuscript too long we suggest moving the data to the supplementary information (suppl.fig 7). Consequently we have merged the original fig. 3c with fig. 4 (new fig. 3) so that auto-transduction is evaluated in the presence or absence of antibiotics. As the data reported in the original

figure 3a and 3b are not central and as we have been pressed for time to complete the needed controls we have not changed the markers in these strains. However we find that our results are absolutely valid as 1) we have obtained the same findings with different strain combinations (suppl. fig 7a + b) and 2) the competition advantage is eliminated by citrate, which demonstrates the involvement of phage (suppl. fig 7). However, as the reviewer points out, these data are not central to the story so another option would be to delete them. We suggest that the editor makes this call.

7. Fig. 3c - legend should be rewritten. Presented results are hard to follow.

Response: The legend to fig. 3c (fig. 3a in the revised manuscript) and the figure was indeed very hard to follow. To improve the figure (and the legend) we removed the CFU's for the individual cultures as they were not contributing any significant information. Further, we have revised the legend to: "Colonies formed on solid medium when at time zero lysogenic 8325-SR (circles) was spotted on agar plates containing a growing USA300 population (squares) forming auto-transduced 8325-SR (triangles) in the absence of antibiotic selection. We find these changes greatly improve the figure and the legend.

8. Fig. 4 and the relevant Result section do not considerably add to the manuscript. The authors may consider transferring them to the Supplementary material if a shortened version is required.

Response: Respectfully we disagree with the reviewer as we find the results shown here are important to illustrate that auto-transduction allows a lysogenic, non-resistant strain to acquire antibiotic resistance from a non-lysogenic, resistant strain even when the latter has the advantage of antibiotics being present in the growth medium. However, to shorten the manuscript while yet highlighting the importance of this finding we have merged the data (original fig. 4) with data from the original fig. 3c to demonstrate that auto-transduction occurs both with (new fig 3a) and without (new fig 3b and 3c) applied selection.

9. Figure 5 - again, controls are missing - lysogen-free bacteria should be added as a recipient in one experiment and also an antibiotic-free bacteria should serve as the donor. These controls must be shown to account for exogenous supply of antibiotic resistance markers from the animal flora, or phages from the animal flora etc.

Response: We acknowledge this important comment and have repeated the experiment while including lysogen-free bacteria as well as antibiotic free donors. The results obtained with these controls have been added to the legend of Fig. 4 replacing the original figure 5.

10. Figure 5 - the legend is unclear. What does it mean n= variable? Which larvae were picked? Which were not? It seems that subjective bias is unavoidable in these experiments.

Response: We have modified the legend (now fig. 4) to clarify the experiment and ease

the reading. To avoid a subjective bias we also monitored transfer in the dead larvae and included the results in the new figure (open circles, figure 4).

11. Figure 5 - perhaps the most interesting finding *in vivo* would be the auto-transduction observed from interactions of the lysogen with the exogenous flora. The experiment as carried out now shows results in larvae that entirely mimic the *in vitro* results, and is thus not highly significant.

Response: The issue of acquisition of resistance genes from endogenous microbiota raised by the reviewer is very interesting. However, the purpose of our experiment was to assess if auto-transduction occurs at *in vivo* conditions and our results demonstrate that this is indeed the case. Yet we anticipate that the broader implications of auto-transduction will be the subject of future studies and that it will inspire other researchers to determine the extent to which it can be used to acquire genes from endogenous microbiota in various hosts.

D. Appropriate use of statistics and treatment of uncertainties

Statistics is not used to indicate p values. These should be critical in Figure 3, and recommended in other figures as well.

Response: We have added statistical analysis in all the figures where it provides a meaningful contribution to the data interpretation. Statistical analysis has been added to fig. 1a, 1c and 1d, fig. 2a and fig. 2b and well as to the supplementary figures 3, 5, 6, and 7. In figure 3a the central finding is that auto-transductants are formed in the absence of antibiotic selection and in figure 3b+c that they are formed even in the presence of increasing concentrations of antibiotics. As the figures show development over time (fig. 3a) or with increasing antibiotic concentrations (fig. 3b and 3c) we believe that the standard deviation is the most meaningful way to display the robustness of the data. Similarly in fig. 4 (originally fig. 5) where we are observing auto-transduction events in *Galleria* at different time points. Again we can see more events at 24 hrs than at 6 hours but the important thing is that they occur and, as displayed now, both in live and dead larvae.

E. Conclusions: robustness, validity, reliability

Conclusions are valid, robust and reliable, provided that the appropriate controls suggested above are implemented and show the expected outcome.

F. Suggested improvements: experiments, data for possible revision

Apart from the above critical points, the experiments are designed to address the issue and provide sufficient ground for the novel conclusions.

G. References: appropriate credit to previous work?

Appropriate credit is given to previous work.

H. Clarity and context: lucidity of abstract/summary, appropriateness of abstract, introduction and conclusions

The manuscript is overall well written and clear. A few comments are listed below.

1. Line 28 - delete "Very".

Response: Has been done, line 29.

2. Lines 64-68, and also 248-254 - the authors term the prophage induction as an altruistic behavior of a subpopulation of the cells. Nevertheless, from the prophage point of view - this is a way to expand its progeny by infecting a new competitor host. Therefore, perhaps the simpler way to evolutionary explain the phenomenon is that the prophage gains advantage rather than the subpopulation of cells is behaving altruistically (a less preferred route of evolution).

Response: We agree with the point raised by the reviewer. We have deleted "altruistic" from line 65, page 4 and have instead added the reviewers point to lines 246-248, page 12 in the discussion

3. Fig. 2 a/b - It is unclear why no transductants are observed in the lysogen-less strain. The frequency of transduction to the lysogenic strain is ~0.1%. One would therefore expect this ratio also in the non-lysogenic strain. The authors indicate that probably no transductants are observed because the non-lysogens are killed, yet show that ~100,000 of them are not killed. This population should yield ~100 transductants. The fact that we do not see it should be explained (perhaps the surviving bacteria are adsorption mutants and therefore cannot be transduced?).

Response: We believe that the reviewer is considering the experiment in the perspective of a normal transduction where lysogens appear and are rescued through inhibition of phage absorption by citrate addition. In our experiment the phage and bacteria interact during the entire 24 hours of the experiment allowing multiple rounds of phage propagation. We acknowledge the point that this was not clearly written in the original manuscript and have clarified this point further in lines 128-134, page 7.

4. Line 155 - rewrite unclear sentence.

Response: This has been re-written for clarifications, line 152, page 8.

5. Lines 160-174 - novelty of the results described in this paragraph is questionable. If manuscript requires shortening - I would recommend transferring it to the supplementary material.

Response: As explained above in the response to comment nr. 6, we agree completely with the reviewer. Therefore, we have chosen to move the original fig. 3a and 3b to supplementary information. This increases the emphasis on the main points of the manuscript and also it shortens the text.

6. Line 192 - remove comma after "sensitive".

Response: This has been done now 185, page 9.

Response to reviewer #2:

This paper reports an interesting finding that builds on decades of phage work: under some conditions, there are some temperate phages (those that can lysogenize their host) that can transduce foreign DNA into a sensitive host, enabling the rapid acquisition of drug resistance (for example). It is generally thought that because transducing particles are known to be minor constituents of a phage population that transduction is a rare event, and because detection of successful transduction usually requires that the phage carrying foreign DNA infect the host largely in isolation of other phages (so that the host cell is not killed).

What they found is that, for lysogens of one temperate phage in *Staph aureus*, the cells readily pick up foreign DNA in a 3 step process: (a) Lysogens 'induce' at some rate, releasing infective phages, which (b) infect sensitive bacteria in their environment. Phage growth and cell lysis releases new phages, some of which had acquired DNA of that sensitive host and are transducing particles. (c) The new phages now infect the original lysogenic strain, which is immune to viable phage infection but is capable of recombining transducing DNA into its chromosome, thereby changing the genotype. It is (c) that is new, albeit primarily in the context of steps of (a) and (b). The study thus provides an interesting mechanism for how transduction can be efficiently observed.

There are a few qualifications that limit the generality of this mechanism. The paper does not delve into any of them.

1) The phage must not only be temperate, but it must be a generalized transducer. A requirement for this is that the phage uses a headful packaging mechanism, which allows non-specific DNA to be packaged into new phage particles. Perhaps half of all phages do not use this mechanism.

Response: We have addressed this point in line 237-240, page 12 by stating that “For auto-transduction to take place the phage must be a generalized transducer and the lysogen susceptible to superinfection. However if only a fraction of the worlds 10^{32} estimated phages harbour these properties auto-transduction is likely a major mechanism in bacterial acquisition of DNA.” In this perspective we find that auto-transduction is still a DNA transfer mechanism of major importance.

2) The lysogen must be susceptible to superinfection, requiring that the receptors are unaltered by lysogenic conversion and that no superinfection exclusion process is conferred by the prophage. Many temperate phages do not have these properties, although the primary phage ($\phi 11$) considered in the study is appropriate.

Response: It is correct that many bacterial taxons express superinfection exclusion

mechanisms that at least in theory could limit to auto-transduction. However, such mechanisms may occasionally be bypassed as shown by for example Ebel-Tsipis & Botstein (Virology 45, p.629-637, 1971) where they demonstrated that superinfection exclusion efficiently inhibits transduction when infecting Salmonella with phage p22 at a multiplicity of infection below 0.5 but at higher multiplicity of infection, infective phage particles overload the superinfection exclusion mechanism and transduction occurs. Further investigations of this issue will be extremely interesting and it presents an excellent objective for follow up studies addressing the generality of the auto-transduction mechanism.

3) The lysogen must sometimes be in an environment with other hosts that are sensitive to the phage (and not already lysogens for that phage). Thus, this mechanism is not likely one for DNA transfer across broad taxonomic levels.

Response: We believe the reviewer intended to write the first sentence as “The lysogen must sometimes be in an environment with hosts that are in-sensitive to the phage”. This is a relevant point. It does not, however, limit the importance of our study as auto-transduction is potentially very important for transfer between closely related *Staphylococcus* taxons (line 225-227, page 11). The importance of horizontal gene transfer for evolution and pathogenicity between close related staphylococci has been demonstrated in several papers from Jodi Lindsay’s group (e.g. Lindsay & Holden, *Funct. Integr. Genomics*. 6:186-201, 2006; Knight et al. *J Antimicrob Chemother*. 67:2514-22, 2012 and Lindsay et al. *Mob. Genet. Elements*. 2:239-243, 2012). Further, the importance of competition between close relatives was emphasized already by Darwin in “The origin of species” where he devoted a subsection of a chapter to the issue entitled “Struggle for life most severe between individuals and varieties of the same species” (pp. 60-63 in our copy, Washington Square Press, Inc, 1963). However, we agree with the reviewer that this is an important an interesting issue and in the future we plan to address the borders of auto-transduction.

The manuscript is very long for its content. In no small part this is because of the simplistic and uneven descriptions of both the formation of transducing particles and of the transduction step. These are both well-established and widely appreciated processes and the text should better reflect what is known and not present material in a classroom style that is more appropriate for beginning students. The manuscript would also benefit from a better use of original references to specific points or of authoritative reviews rather than the plethora of secondary or derivative publications currently cited.

Response: We acknowledge the point made by the reviewer and have carefully edited the manuscript to remove sections that may appear textbook-like. In consequence we have shortened:

- Lines 108-110 in original manuscript are now just line 111
- Lines 110-112 in original manuscript are reduced to line 112
- Lines 112-115 in original manuscript are reduced to 113-115
- Lines 116-117 in original manuscript are deleted
- Lines 123-131 in original manuscript are deleted and the relevant information

about citrate and phage immunity was added to the following paragraph

- Lines 160-174 in original manuscript are reduced to 158-167.
- Lines 180-181 in original manuscript are deleted
- Lines 185-187 in original manuscript are deleted
- Also we have moved the original fig. 3a and 3b to supplementary information as suggested by reviewer 1. Lastly we have changed the citations used in the introduction to reduce the number (from 56 to 47) and are using authoritative reviews where appropriate.

Throughout the manuscript the authors must clearly distinguish the efficiency of autotransduction and transduction from the efficiency of observing or detecting transductants - e.g., only the latter is 10^6 more efficient (line 142).

Response: We believe that the reviewer is suggesting that we should consider how we define the efficacy. Normally transduction efficiency is determined as obtained colonies relative to the number of phages added to the experiment. However, as we during auto-transduction have continuous production of phages and multiple rounds of infection we have assessed the efficiency of transductions and auto-transduction as relative to the total number of bacteria in the experiment.

Minor comments:

19. replace "dormant" with "quiescent", has been replaced, line 20.

25. "so-called" is a completely inappropriate term, has been deleted, line 26

75. are phi12 and phi13 also capable of transduction? Do they affect any of the data shown (it is appreciated that some experiments were done with phi11 alone).

Response: Both phi12 and phi13 are *cos* phages and therefore they are not capable of generalized transduction (Chen et al., ISME J. 9(5):1260-3, 2015). Further neither phi12 nor phi13 are released during growth (results obtained in our laboratory) and therefore they will not affect the data obtained.

244-5. P22, the phage used by Zinder and Lederberg, is now understood to confer both superinfection exclusion and lysogenic conversion when present as a prophage.

Response: We acknowledge the point raised by the reviewer. However as Zinder and Lederberg, already in their early experiments, observed a phenomenon we interpret as auto-transduction mediated by P22 it is likely that the superinfection exclusion mechanism can be overridden possibly by high phage numbers as shown by Ebel-Tsipis & Botstein (Virology 45, p.629-637, 1971).

Response to reviewer #3:

Introduction

The manuscript by Haaber et al. presents data supporting the conclusion that release of phage from bacterial cells allows viable prophage-containing cells to acquire antibiotic-resistance genes released from phage-susceptible strains. The authors term this process

'auto-transduction' and suggest that this process could explain why *S. aureus* can rapidly acquire antibiotic resistance.

General Points for Consideration

The work is of interest to those working in the area of antibiotic resistance and represents a novel finding of significance. The data is of a high standard and the appropriate statistics have been applied. However, the manuscript is difficult to read and understand. The authors would be advised to re-write the Results section to facilitate better understanding by researchers that are not familiar with the concepts of phage biology. Also, the Figure legends are poor and should include more and better explanation of the presented data.

Response: We acknowledge the comments by the reviewer and have in response edited the text to make the manuscript easier to understand. More specifically we have added additional information in lines line 75-76, line 99-102, line 130-136 and line 251-254. Further we are thankful for the comment about the figure legends as they were indeed hard to follow. We have now carefully edited the figure legends so that they are more comprehensive and provide better explanation of the experiments. Also we have in fig. 3a removed information about the growth of individual cultures from the figure to enhance clarity.

It would be beneficial to the reader if the Supplementary Table 2, listing the strains that were used, was in fact included in the actual manuscript to aid reader understanding.

Response: Since the second reviewer expresses concerns about the length of the manuscript we have kept the strains in the supplementary information as a table but we will be very happy to include for example the first 15 strains most commonly used in the manuscript in a table within the manuscript while leaving the remainder in the supplementary information. We believe that the decision is best left with the editor in case the manuscript is accepted.

Specific Points for Consideration

In all co-culture experiments with different *S. aureus* strains there appears to be no consideration of the possibility that different strains could have different growth rates. If this were the case have the authors measured this? In competition experiments how would different growth rates resulting in potentially widely different proportions of the different strains effect results in terms of frequency of resistance, CFU or auto-transduction? This must be addressed.

Response: The growth rates of the relevant bacterial strains have now been examined and added as Supplementary Table 4. In most cases, the growth rates were not statistically different but in a few cases, the non-lysogenic strains would grow faster than the lysogenic strains against which they compete (possibly due to the loss of lysogenic cells caused by the phage release). However, we believe that the fact that our results show that the phages help the lysogenic strain to out-compete the non-lysogenic competitor despite a lower growth rate of the lysogenic strain just accentuates our

point of phages being a competitive asset of the lysogenic strain.

Throughout the manuscript relative cell numbers are expressed as OD600 values. This is an inaccurate measure of cell numbers. The size of bacterial populations should be expressed as CFU/mL.

Response: We believe that it has escaped the attention of the reviewer that we do indeed provide all our results as CFU/ml through the manuscript.

Line 36 - the statement that MRSA is spreading epidemically is overstated. For example, incidence of MRSA in UK hospitals is in decline over the past few years due to effective interventions to break the chain of infection.

Response: We acknowledge this point and have modified the statement, line 37, to saying "(MRSA) strains have in recent years spread"

Line 75 - explain the significance to the reader of the mutation in *rpsL*.

Response: The significance of the *rpsL* mutation in yielding streptomycin resistance has now been clarified, line 75, page 4.

Line 82 - the lysogenic strain can be distinguished from the non-lysogen because it is non-haemolytic due to increased activity of SigB and SarS. Referring to the previous comment above could the differential activity of these important regulatory proteins effect growth rate and thus results in competition experiments? could they effect intrinsic antibiotic resistance? etc. Has this been measured?

Response: We have measured the growth of all relevant strains (added as Suppl. Table 4) and find that in most cases they grow with the same growth rate (see also response above). We have also assessed the intrinsic resistance of the strains and found that they vary maximally two fold. The latter point is, however, in our opinion of less importance as antibiotic selection is not applied in most experiments and in the experiment shown in the new fig. 3 we only apply the antibiotics that select for the transferred resistance. The minimal inhibitory concentration for these strains have been added in the methods section lines 350-351, page 18.

Line 85-100 needs to be rewritten. It is difficult to understand and needs to link with Figure 1 more effectively. For example, the reader is referred to Fig 1a in line 89 but there is no explanation of what strain 8325-SR is until line 93.

Response: We acknowledge introduce a number of strains and that the reader will be directed to fig. 1 before the strain 8324-SR has been described. However, after careful consideration we find that the sequential introduction of strains is most pedagogical and communicates in the best way the strain composition and properties. Therefore, we suggest that the introduction of strains is not changed.

In Figure 1, why is the frequency of double-resistant bacteria identical to triple-resistant bacteria for both 8325-S and 8325-SR? Explain.

Response: With our experimental setup, we observed, as expected, that traditional

transduction of the non-lysogen did not occur. Also strains 8325-S and 8325-SR grow with equal growth rates. Therefore we predict that 8325-S and 8325-SR will be equally able to attract antibiotic resistance genes by auto-transduction and this is indeed what we observed.

Line 100 - after the sentence insert data not shown.

Response: We have inserted the data providing the detection limit in the legend to fig. 1a.

Line 283 - This sentence must be re-written for understanding.

Response: We have re-written the sentence to explain that auto-transduction can be calculated by dividing the number of cells carrying all the antibiotic markers that were carried separately by the co-cultured lysogen and non-lysogenic strains with the total number of CFU in the co-culture. The sentence now reads (line 284-286, page 15): "Auto-transduction frequencies were determined as the fraction of cells in a co-culture that were resistant to the antibiotic markers carried by both of the co-cultured strains relative to the total number of CFU"

Line 317 - The sentence beginning 'In the absence..' needs to be re-written with a full explanation for understanding.

Response: We have re-written the sentence (line 322-324, page 16): "Other phages released from USA300 or Newman were enumerated by flow cytometry (Becton Dickinson FACS Calibur) as previously described"

Line 320 - define abbreviations FSC, SSC.

Response: modified accordingly, line 323-324, page 16.

The data presented in Figure 5 is inadequately presented and explained. The authors state that a 'lower inoculum' was used to prevent rapid killing. They then state that an inoculum of 10^6 was used - this seems high? The authors need to present the number of larvae that died out of the 20 inoculated in Fig 5a and b. In Fig 5a there are a total of 19 data points from each larva suggesting only 1 died? In Fig 5b there are 10 suggesting 10 died? Because the authors only sampled living larvae would this disparity in the number of surviving larvae not skew the auto-transduction calculation - in fact, why not sample bacteria from all infected larvae alive or dead?

Response: 10^6 may indeed appear as a high number for an inoculum. It is, however, chosen because the larvae at this inoculum of staphylococcal cells demonstrate 75% survival after 48h as shown by Desbois & Cooté (J Antimicrob Chemother. 66:1785-90, 2011). In order not to confuse the reader we have deleted the notion of "lower" from the text (line 357, page 18). Regarding the issue of presenting results of both dead and live larvae we have edited Figure 5 (now fig. 4) so that it now present both categories.

Line 359 - how was the 'multi-dispenser' used with the Hamilton syringe to infect larvae? Explain.

Response: we have added the description to Materials and Methods section, line 360-362.

Further changes:

In addition to addressing the issues raised by the reviewers we have changed the presentation of the data in figure 2 to highlight the central findings obtained in the experiment.

REVIEWERS' COMMENTS:

Reviewer #1 (Remarks to the Author):

The authors have adequately addressed my concerns.

Reviewer #3 (Remarks to the Author):

[No further comments for author, he/she recommends publication.]